# Effects of Dietary Supplementation of L-Carnitine and Mannan-Oligosaccharides on Growth Performance, Selected Carcass Traits, Content of Basic and Mineral Components in Liver and Muscle Tissues, and Bone Quality in Turkeys

**DOI:** 10.3390/ani13040770

**Published:** 2023-02-20

**Authors:** Małgorzata Kwiecień, Karolina Jachimowicz-Rogowska, Wanda Krupa, Anna Winiarska-Mieczan, Magdalena Krauze

**Affiliations:** 1Department of Animal Nutrition, Institute of Animal Nutrition and Bromatology, University of Life Sciences in Lublin, Akademicka Str. 13, 20-950 Lublin, Poland; 2Department of Bromatology and Food Physiology, Institute of Animal Nutrition and Bromatology, University of Life Sciences in Lublin, Akademicka Str. 13, 20-950 Lublin, Poland; 3Department of Animal Ethology and Wildlife Management, University of Life Sciences in Lublin, Akademicka Str. 13, 20-950 Lublin, Poland; 4Department of Biochemistry and Toxicology, University of Life Sciences in Lublin, Akademicka Str. 13, 20-950 Lublin, Poland

**Keywords:** L-carnitine, mannan-oligosaccharide, turkeys, poultry, growth performance, carcass traits, mineral composition, bone quality

## Abstract

**Simple Summary:**

L-carnitine and mannan-oligosaccharides (Bio-Mos) are nutritionally and nutraceutically important additives used as dietary supplements to improve production performance, carcass quality, and feed quality. In this study, the effect of application of L-carnitine and Bio-Mos on selected production parameters and content of basic and mineral components in turkey liver, breast, and thigh muscles and on the physical, morphometric, strength, and mineral composition parameters of femurs was examined. The use of both additives was found to result in a significant increase in the body weight of the birds in the final fattening period. Concurrently, it reduced the amount of fat in the breast muscles and liver. The turkey femurs were also characterized by higher elasticity and breaking strength values. The results indicate the need for further research to determine the optimal doses of L-carnitine and Bio-Mos to be used in poultry nutrition.

**Abstract:**

The study aimed to determine the effect of L-carnitine and Bio-Mos administration on selected production performance, slaughter parameters, elemental and mineral content of liver, breast and thigh muscles, and physical, morphometric, strength and bone mineral composition parameters of turkeys. The experiment was conducted on 360 six-week-old Big-6 turkey females, randomly divided into three groups of 120 birds each (six replicates of 20 birds). The turkeys of the control group were fed standard feed without additives; group II was fed with drinking water, a preparation containing L-carnitine at a dose of 0.83 mL/L, while group III was provided mixed feed with 0.5% Bio-Mos. The addition of L-carnitine and Bio-Mos increased body weight at 16 weeks (*p* = 0.047) and reduced the proportion of fat in the breast muscle (*p* = 0.029) and liver (*p* = 0.027). It also modified the content of some minerals in breast muscle, thigh muscle, liver, and bone. Furthermore, the addition of L-carnitine and Bio-Mos increased bone mass and length and modified the value of selected morphometric and strength parameters. The results indicate a positive effect of the applied feed additives on selected rearing indices and carcass quality while improving the elasticity and fracture toughness of the femur. There is a need for further research to determine optimal doses of L-carnitine and Bio-Mos in poultry nutrition.

## 1. Introduction

A systematic increase in the consumption of poultry meat has been observed in recent years [1], and FAOSTAT forecasts indicate that this trend will continue [2]. Therefore, methods for improvement of meat quality indicators are constantly being developed. Carcasses of modern broilers are characterized by e.g., an excess amount of fat, which is unfavorable from the consumer’s point of view; this can undoubtedly be ascribed to genetics and ingredients supplied with feed [3]. It is therefore necessary to improve the carcass composition, which can be achieved by the use of various feed additives. Additives with immunomodulatory properties seem to be valuable tools for improvement of feed conversion rates, growth rates, and breast muscle weight, and for the reduction of the body fat level. Some prebiotics, in particular mannan-oligosaccharides (MOS; Bio-Mos) or L-carnitine, seem to meet these expectations [4,5,6,7,8]. MOS are commonly used in poultry nutrition [9] mainly due to their beneficial effects on the gut health and growth of birds [10,11]. The addition of prebiotics has also been studied as a potential alternative to antibiotic growth promoters [12,13,14]. The beneficial effect of these additives is associated with their properties.

MOS are modulators of the intestinal microbiome, which plays an important role in the stimulation of defense mechanisms in the organism, activates digestive enzymes, enhances nutrient absorption from the feed through elimination of pathogenic bacteria, and neutralizes toxins secreted by pathogens [15]. The positive effects of this compound on poultry include improvement of slaughter performance and reduction of susceptibility to digestive and respiratory infections [11]. MOS, which are not digested in the stomach, remain unchanged in the small intestine, where they stimulate the intestinal flora activity [16]. Prebiotics improve the absorption of macro and micronutrients, thereby increasing their availability to be utilized by the organism [17].

L-carnitine seems to be another alternative for achievement of high health indices [18]; this additive improves the quality of meat and exhibits potential ergogenic properties [19]. It is synthesized mainly in the liver and in the brain and kidneys. It plays an essential role in lipid metabolism via transport of triacylglycerols from the cytoplasm to the mitochondrion. In the mitochondrion, triacylglycerols are oxidized in the β-oxidation process resulting in the production of energy in the form of ATP (adenosine triphosphate) [20]. L-carnitine concentrations in animals vary depending on the tissue type [21], animal species [22], animal nutritional status [23], and feed quality [24]. Lysine and methionine are the primary precursors of L-carnitine in the organism; therefore, an optimal supply and intake of these amino acids prevents its deficiency. In some situations, e.g., an L-carnitine-low diet, stress conditions, a fat-low diet, and limited biosynthesis of L-carnitine, the compound becomes an essential nutrient in young animals [25]. The content of L-carnitine in feed is quite low, and products of animal origin are its main source. A limited level of methionine and lysine results in its deficiency. L-carnitine deficiency inhibits the metabolism of fatty acids and intensifies glycolysis, which reduces glycogen reserves. In such a case, the synthetic form of the compound, which is actively absorbed in the small intestine, can be supplemented [26].

An important issue in modern poultry farming systems is the proper functioning of bones, as it affects growth and metabolism in the organism, and problems with legs in turkeys are one of the main factors limiting the profitability of production of these birds [27]. Bones, mainly femurs and tibias with their joints, are subject to excessive loads caused by the hypertrophy of breast muscles and the shift of the center of gravity affecting body balance [28]. Bones in modern broiler lines are characterized by lower calcification and high porosity, which can lead to bone damage [29]. Disturbances in the proper development of leg bones in poultry are attributed to many factors; however, optimization of nutrition can significantly reduce their frequency [30]. As shown in research reports, L-carnitine supplementation can improve bone density and microstructural properties [31,32,33]. Studies conducted in rat models have demonstrated a beneficial effect of prebiotic supplements on the absorption of minerals [34], bone mineralization [35], or bone architecture measured as the femur volume [36]. As highlighted by some authors [37], the effect of various fructooligosaccharides on Ca bioavailability and bone mineralization is not uniform and depends on the degree of their polymerization.

The combination of Bio-Mos and L-carnitine would make this combination much more beneficial than the application of single ingredients. Bio-Mos is a prebiotic, and therefore it stimulates the development of a beneficial intestinal microbiome, improves the morphological structure of the intestinal epithelium, protects enterocytes, and also stimulates the immune system located largely in the digestive tract. On the other hand, L-carnitine facilitates the penetration of fatty acids into cells, thanks to which they can be used to improve cell energy and reduce glycogen and fat reserves in the carcass. This entails a measurable health effect and improves weight gain. In addition, L-carnitine stimulates the distribution of important short-chain fatty acids, e.g., butyric acid, which is a product in the enterocytes of the intestinal microbiota. This, in turn, can stimulate the work of intestinal enteroendocrine cells and regulate their hormonal activity [38,39,40].

While the effect of L-carnitine and Bio-Mos supplements on production results (increase in body weight, weight gains, increased feed conversion and intake) is well documented in the scientific literature [5,6,25,41,42], their impact on the composition of the body, some organs, and bone quality is still insufficiently known. This is particularly important for the production of good-quality meat with optimal nutritional value for consumers and for increasing the profitability of turkey breeding and rearing. It was therefore assumed that the application of an appropriate dose of L-carnitine and Bio-Mos contributed to improvement of e.g., the mineral status of the organism, rearing indicators, and bone quality.

The present study was focused on analyses of the effect of L-carnitine and Bio-Mos supplementation on selected production indices, carcass parameters, and the content of basic and mineral components in the liver, breast muscles, and thigh muscles as well as physical, morphometric, and strength parameters and bone mineral composition in turkey females.

## 2. Materials and Methods

### 2.1. Bird, Experimental Design, and Management

All procedures used throughout this study were approved by the Local Animal Welfare Committee at the University of Life Sciences in Lublin, Poland (Resolution No. 59/2010).

The study involved 360 Big-6 turkey females. Six-week-old birds were purchased and assigned randomly into three experimental groups with 120 birds each (20 per subgroup and 6 replicates per subgroup). The birds were kept on straw bedding in pens (3.7 × 4.15 m providing 0.384 m^2^/bird) until 16 weeks of age. The stocking density at the initial stage of rearing was 4.5 birds/m^2^. The birds were kept in the same zoohygienic conditions optimal for turkey fattening. They were reared under constant veterinary supervision. Standard complete mixtures for turkeys were used in the experiment. The content of basic nutrients in the feed mixtures (Table 1) was consistent with the NRC recommendations [43]. Throughout the experiment, the turkey hens had constant access to drinking water and received standard full-portion granulated mixtures *ad libitum* (Table 1). From the 6th week of the experiment, feeds with the composition shown in Table 1 were used. The light schedule provided 17 h of light with the light intensity of 5 lux until day 5 and 5 lux thereafter. The temperature from the 42nd day of rearing was 15 °C. The temperature and lighting programs were consistent with the recommendations of British United Turkeys [44]. Table 2 presents the analyzed macro and micronutrient content in the feed mixtures provided in each rearing period. The Ca, Mg, Cu, Fe, Zn, and K content in the feed was measured using flame atomic absorption spectrophotometry (FAAS) (Unicam 939/959AA-6300, Shimadzu Corp., Tokyo, Japan). The total P content in the feed was determined according to the Polish standard PN-76/R-64781 using a Helios Alpha UV-VIS spectrophotometer (Spectronic Unicam, Leeds, UK).

The addition of L-carnitine or Bio-Mos was the experimental factor. The experimental design is shown in Table 3. Group I was the control group that did not receive L-carnitine or Bio-Mos. The birds in group II received a commercial formulation for poultry containing L-carnitine (30 g L^−1^) in a dose of 0.83 mL L^−1^ (Biofaktor, Poland) with drinking water, while the turkey females in group III were given a feed mixture supplemented with 0.5% of Bio-Mos (Alltech, Poland).

### 2.2. Experimental Measurements

Ten hours before slaughter, the turkeys were not fed but had constant access to water. In the 16th week of rearing, the turkeys were weighed in the morning before the slaughter and two birds (12 birds per group) with representative body weight were selected from each group. A 3-phase feeding programme (week 6–9, 10–13, and 14–16) was used during the study. At 6, 9, 13, and 16 week the turkey hens were weighed and feed intake was recorded. The body weight gain of the bird sand feed conversion ratio was calculated for each feeding period. Mortality rates were recorded daily and the body weights of all dead birds were used to adjust for average daily gain, average daily feed intake, and feed conversion ratio. The post-slaughter weight of turkey females was included in the calculation of the FCR. The birds were euthanized after electrical stunning, hung on a processing line, and exsanguinated for 3 min by a unilateral neck cut severing the right carotid artery and jugular vein. The stunning and slaughter of the turkeys was carried out in accordance with the guidelines recommended by Council Regulation (EC) No 1099/2009, 24 September 2009, the title of the procedure is: On the protection of animas at the time of killing, Official Journal L, p. 303, 2009. These were procedures respected and used by the plant where the experiment was conducted. After earlier scalding at 61 °C for 60 s and defeathering in a rotary drum picker for 25 s, a simplified carcass analysis [45] was performed during dissection, and liver tissue, breast and thigh muscles, and femoral bones were collected.

The skin was separated from the muscles, and the muscles and liver were packed in individual sealed plastic bags, labeled, and kept frozen at −80 °C until chemical analyses.

### 2.3. Sample Collection and Chemical Analyses

#### 2.3.1. Diets and Muscles

The content of dry matter (Method 925.09), crude ash (Method 923.03), crude protein (Method 920.87), and ether extract (Method 920.39) was determined in the muscles and livers (n = 36) using AOAC [46].

The Ca, Mg, Cu, Fe, Zn, K, and Na content in the muscles and livers was measured using flame atomic absorption spectrophotometry (FAAS) (Unicam 939/959AA-6300, Shimadzu Corp., Tokyo, Japan). Approximately 1-g aliquots of breast, thigh, and liver tissues were placed in pre-ignited porcelain crucibles. Next, the samples were incinerated in a muffle furnace at 550 °C for 24 h and the ash was dissolved in 10 mL of 1M HNO_3_. The content of minerals was determined as follows: Ca at λ = 422.7 nm, Mg at λ = 285.2 nm, Cu at λ = 324.8 nm, Fe at λ = 248.3 nm, Zn at λ = 213.9 nm, Na at λ = 589 nm, and K at λ = 766.5 nm [47]. The total P content in the feed was determined according to the Polish standard PN-76/R-64781 [48] using a Helios Alpha UV-VIS spectrophotometer (Spectronic Unicam, Leeds, UK). The certified reference material (BCR-063R) contained: Mg 1269 g‧kg^−1^, Ca 13.49 g‧kg^−1^, K 17.68 g‧kg^−1^, Zn 49 mg‧kg^−1^, Cu 0. 602 mg‧kg^−1^, Fe 2.32 mg‧kg^−1^, and P 11.1 g‧kg^−1^.

#### 2.3.2. Bone Analyses

Dissected bones were weighed and the length (using an electronic caliper; 0.001 mm accuracy) and circumference in ½ of the bone length were measured. Based on these measurements, the bone density index (BI) was calculated as a ratio of the bone weight (in mg) to the length (in mm) [49]. The mechanical properties of the bones were determined in the three-point bone bending test performed with the use of the Zwick Z010 apparatus (Zwick GmbH & Co KG, Ulm, Germany) equipped with a measuring head (Zwick GmbH & Co KG) and connected to a computer with TestXpert II 3.1 software (Zwick GmbH & Company KG). The head worked at a constant speed of 10 mm‧min^−1^ [50,51]. The following were the bone strength parameters: yielding deformation (dy), bending point resistance (Wf/A), and load-to-deformation ratio (Wy/dy) were determined on the basis of the maximum elastic force (Wy) and yielding load (Wf) [50,51,52]. The external and internal horizontal and vertical diameters of the bone shaft cross-section were measured to determine the following geometrical indices: second moment of inertia (Ix), cross-sectional area (A), mean relative wall thickness (MRWT) and cortical femoral shaft indices: cortical indices (CS), cortical layer thickness (CLT), cortical surface (CS), and cortical index (CI) [51,52]. Next, the femurs were degreased, dried to constant weight, and mineralized. The content of Ca, Mg, Zn, Cu, K, and Fe was determined in an ASA SOLAR 939 UNICAM flame spectrophotometer [47]; the phosphorus content was determined in an Spectronic Helios Delta 9423 UVD flame spectrophotometer (Wilmington, North Carolina, USA) [48]. The content of selected mineral components in the bone was expressed as the content of these components in crude ash.

### 2.4. Statistical Analysis

The numerical data were processed with the one-way ANOVA analysis of variance (α = 95%; *p* < 0.05). Their arithmetic mean, standard error of the mean (SEM), and significance level (*p*) were specified. The significance of differences between the mean values of the analyzed parameters was determined using the Duncan test (post-hoc test) in the Statistica 10.0 program.

## 3. Results and Discussion

### 3.1. Growth Performance and Selected Carcass Characteristics

The nutrition for animals with a high production potential should ensure maximum production performance and maintain good health of animals through its beneficial effect on the gastrointestinal tract, metabolism, and stimulation of the immune system. Hence, various feed additives are used in mixtures used in poultry nutrition, e.g., prebiotics, which inhibit the growth of intestinal pathogens concurrently replacing antibiotics, whose use is now prohibited [42,53]. Investigations focused on determination of the optimal level of application of prebiotics bring health benefits to animals [54] and contribute to the improvement of production indices and the profitability of rearing [55]. Dietary L-carnitine supplementation may additionally exert a positive effect on the chemical composition of carcasses [19,56] and, in combination with recommended levels of lysine and methionine, may improve the immune response in the organism [18].

Table 4 shows the effect of L-carnitine and Bio-Mos on the production results and selected carcass parameters (liver, breast muscle, and thigh muscle weight). The results of other studies on the addition of L-carnitine and Bio-Mos on production performance are ambiguous. However, the present study showed a 5.8% and 5.7% increase (*p* = 0.047) in the body weight of turkeys receiving both L-carnitine and Bio-Mos, respectively, in the sixteenth week of rearing, compared to the control group. Similarly, as reported by Ognik and Krauze [57], the addition of Bio-Mos in the dose of 0.5% per kg of feed increased the body weight of hens in the final rearing period by approx. 6%, compared to the non-supplemented control group. In another study [58], Bio-Mos supplementation at a level between 0.5 and 2 kg/ton of feed resulted in an approx. 2% increase in the body weight of birds, compared with the control. Furthermore, Silva et al. [59] and Haldar et al. [60] reported improved growth rates in broiler chickens fed with Bio-Mos. A study conducted by Teng et al. [42] demonstrated an increase in body weight (by 2.8%) and weight gain (by 3.1%) of broiler chickens receiving Bio-Mos during the first 2 weeks of rearing. As reported by Ghazalah et al. [41], a group of chickens fed a basic diet supplemented with 2 g MOS/kg exhibited a significant increase in body weight compared to groups receiving 0.5 g and 1 g/kg Bio-Mos and the control group in all rearing periods. Better production parameters (body weight, body weight gain) after Bio-Mos supplementation compared to the control group were also obtained by Nikpiran et al. [61], Tufail et al. [62], and Rehman et al. [63]. These results confirm that mannans can be an effective alternative to antibiotic growth promoters in animal feed, as they exert a positive effect on the microbiome and nutrient absorption [6,64]. In contrast to the results obtained in the present study, El-Saway et al. [65] and Thabet et al. [66] found that L-carnitine supplemented at the dose of 0.25 g/L and 150 mg/kg feed had no effect on the weight gain in broiler chickens. In turn, the use of L-carnitine in higher amounts (i.e., at the level of 300–800 mg/kg) resulted in an increase in body weight compared to the control group [5]. As shown by Khoshkhoo et al. [67], L-carnitine can significantly improve weight gain only between the 35th and 49th day of broiler chicken rearing, with no effect in earlier periods. Hossininezhad et al. [68] reported a significant decrease in FCR throughout the L-carnitine administration period. In contrast, Kidd et al. [69] observed no effect of L-carnitine supplementation on broiler performance. It is likely that the weight gain in broiler chickens receiving L-carnitine is related to the role of this compound in the enhancement of the oxidation of long-chain fatty acids and in increasing the amount of acetyl-coenzyme A in the mitochondria, which intensifies the utilization of dietary protein in poultry [70]. It is possible that the effect of L-carnitine depends on the content of lysine and methionine in feed [18].

In the present study, the addition of L-carnitine and Bio-Mos had no effect on the weight of breast and thigh muscles and the liver (Table 4). Similar results were obtained by Fujimoto et al. [71] and Murali et al. [3], who showed no significant effect of L-carnitine supplementation on broiler carcass parameters. In their study with L-carnitine addition, Ghoreyshi et al. [18] did not show a significant effect of the compound on the breast muscle and liver weight; however, they reported a significant reduction in e.g., thigh muscle weight. In another study, the use of 100 and 50 mg/kg L-carnitine in broiler diets contributed to an increase in the breast weight but did not exert a significant effect on the offal weight [72]. In a study conducted by Khemalapure et al. [73], higher liver weight indicators were observed in chickens receiving feed with the addition of 0.05% and 0.075% Bio-Mos; however, the results were not statistically confirmed. Moilwa et al. [74] found significant differences in the chicken liver weight between the experimental groups (1 g/kg, 2.5 g/kg, and 5 g/kg Bio-Mos supplementation) and in comparison to the control group. The highest liver weight was recorded in chickens receiving the higher doses of the prebiotic in the diet (2.5 g/kg and 5 g/kg), whereas the lowest value of this parameter was noted in the group supplemented with the lower prebiotic dose (1 g/kg). In turn, in a study conducted by Yang et al. [75], the use of Bio-Mos in the diet for broilers reduced the liver weight, which was probably related to the supporting role of L-carnitine in the oxidation of lipids contained in the feed and prevention of the accumulation of fats in the liver. Additionally, L-carnitine produced in the liver is transferred to muscle tissue. The majority of L-carnitine, which is a mixture of L-carnitine synthesized in the organism and that absorbed with the feed, is stored in skeletal and cardiac muscles. The use of L-carnitine in poultry feed contributes to an increase in energy efficiency, which helps poultry to utilize energy from feed lipids faster and more easily [76].

### 3.2. Content of Basic and Mineral Components in the Breast and Thigh Muscle

The results of the analysis of the basic components in the breast muscles (Table 5) showed that the addition of both Bio-Mos and L-carnitine reduced (*p* = 0.043) the crude ash content by 14.9% and 10.7%, respectively, compared to the control group. Similar results were reported by Mehdizadeh Taklimi et al. [76], who used powdered L-carnitine in the diet and observed a significant decline in the content of crude ash in the breast muscle. In turn, in a study on chickens, El-kelawy [77] did not show any significant differences (*p* = 0.658) in the crude ash content between birds supplemented with different doses of L-carnitine in the feed and the control group. In the present study, there was no effect of the L-carnitine supplementation on the crude protein content in the breast muscle. Similar results were obtained by Mehdizadeh Taklimi et al. [76]. In a study conducted by Dev et al. [78], the crude protein content in the breast muscles of broiler chickens increased in groups receiving feed supplemented with 0.2% Bio-Mos and lactic acid bacteria in the amount of 10^6^ CFU/g of feed and 10^7^ CFU/g of feed (25%), compared to the control group (22.4%) (*p* < 0.05). In turn, the present study showed no differences in the crude protein content between the groups.

In addition to the crude ash content, significant (*p* < 0.05) differences were observed in the crude fat content in the breast muscle (Table 5). The breast muscles of turkeys receiving Bio-Mos and L-carnitine contained 23.7% and 25.8% lower levels of crude fat, respectively, compared to the control birds. Similarly, Bonos et al. [79] and Biswas et al. [64] showed that Bio-Mos inclusion in the diet reduced the crude fat content by 52.6% in quail breast muscle and by 23.2% in chicken breast meat, respectively. The Bio-Mos additive used in a study conducted by Lipiński et al. [80] resulted in a decrease in the content of this component. Chickens fed with diets supplemented with 50 mg/kg L-carnitine were characterized by the greatest reduction (*p* = 0.007) in the fat content (by 32.9%) in the breast muscles in comparison with birds supplemented with higher L-carnitine doses and with the control group [77]. L-carnitine has been shown to be effective in reducing the fat content in various carcass parts, including thigh and breast muscles [5,81]. Different results of the crude fat content were obtained by Xu et al. [82]. The decrease in the crude fat content in birds fed with L-carnitine is probably related to its effect on fat catabolism in the organism, which results in reduced carcass fat deposition [83,84,85]. This may be caused by a decrease in the activity of CPT-I carnitine transferase and a consequent decrease in β-oxidation of fatty acids.

In the present study, significant (*p* < 0.05) differences between the experimental groups were observed as well. The Bio-Mos-supplemented group exhibited 2.8% higher fat content than the group receiving L-carnitine. The present study showed that, compared to the control group, the supplementation with Bio-Mos had an effect on the content of crude ash and fat in the breast muscle. This finding differs from the data reported by Cheng et al. [86], who showed that Bio-Mos inclusion had no effect on the chemical composition in the breast muscle.

The Bio-Mos supplementation produced a significant (*p* < 0.05) increase (by 10.4%) in the Na content in the breast muscle, compared to the group receiving the L-carnitine-supplemented feed (Table 5). Moreover, significantly (*p* < 0.05) higher (by 36.4%) Cu content was found in the Bio-Mos-supplemented group than in the control. The present results may be associated with the abundance of minerals in yeast [17]. There were no significant differences between the levels of Mg, Ca, K, Fe, and Zn in the turkey breast muscles. In turn, Roberfroid [87] showed that mannans had an impact on the effectiveness of Mg absorption and increased its content in muscles.

Table 6 shows the content of basic nutrients and minerals in the turkey thigh muscles. The increase in the content of basic components in the leg muscles (not confirmed statistically) may be associated with an increase in tissue weight. There were no significant differences in the content of dry matter, crude ash, crude protein, and crude fat between the groups. Abdel Magied et al. [88] observed a significant increase in crude protein in all experimental groups (supplementation of the basic diet with 0.3% of turmeric, 0.05% of Bio-Mos, 0.015% addition of Biostrong 510, 0.05% of Bio-Mos + 0.3% of turmeric, and 0.05% of Bio-Mos + 0.015% of Biostrong 510) compared to the control. Similarly, Dev et al. [78] reported a significant (*p* < 0.05) increase in the crude protein content in thigh muscles of broiler chickens receiving a diet supplemented with 0.2% of Bio-Mos and lactic acid bacteria in the dose of 10^6^ CFU/g of feed and 10^7^ CFU/g of feed (15.9%), compared to the control group (13.9%). In turn, Corduk et al. [89] observed a significant increase in the content of dry matter in the tested muscles in a study on the effect of L-carnitine supplementation.

The content of minerals in muscles determines their quality and technological value. It also has an impact on the dietary value of carcass parts and helps to meet consumer expectations [90]. The factors applied in the study had various effects on the concentration of macro and micronutrients in the soft tissues (Table 6). There were significant differences in the content of Ca, Fe, and Cu between the groups. After 16 weeks of the experiment, the highest (0.039 g·kg^−1^) concentration of Ca in the thigh muscles was observed in the L-carnitine-supplemented group, and the lowest value (0.032 g·kg^−1^) was recorded in turkeys receiving standard feed. Different results were reported by Oliveira et al. [91], who found that mannans affected Ca retention in the organism. The addition of Bio-Mos increased the Fe content by 20.2% compared to the control and by 36.1% compared to the L-carnitine-supplemented group. The Cu content differed significantly (*p* < 0.05) between all groups. Its highest content was detected in the group receiving the L-carnitine-supplemented diet; it was 35.7% higher than in the control group and 69.3% higher than in the Bio-Mos-supplemented birds. The Bio-Mos addition also increased (*p* = 0.044) the content of Cu (by 52.3%) in the thigh muscle, compared to the control. Kidd et al. [69] reported an increase in the weight of thigh and drumstick muscles in broilers supplemented with L-carnitine (*p* < 0.05). The results may also be related to the effect of L-carnitine on breast muscles [92] and the abundance of minerals in yeast [17].

### 3.3. Content of Basic and Mineral Components in the Liver

Table 7 shows the content of basic nutrients and minerals in the turkey livers. The addition of both L-carnitine and Bio-Mos caused a 22.6% and 12.9% decrease in the crude fat content in the liver, respectively, compared to the control group. This was probably associated with the ability of both pro-health preparations to reduce the organ fat level, which was also reported by Xu et al. [82]. There were also differences in the fat content within the treatment groups; the Bio-Mos-supplemented group exhibited approximately 11% higher fat content in the liver than the birds receiving L-carnitine.

The addition of Bio-Mos significantly (*p* = 0.037) decreased (by 60%) the content of Ca in the turkey liver tissue compared to the control group (Table 7). Different results were obtained by Oliveira et al. [91], who found a positive effect of mannans on Ca retention in the organism. In the present study, the highest Fe content was detected in the livers of turkey females receiving L-carnitine, and the difference (27.5%) was significant in comparison with the control group. The supplementation with Bio-Mos and L-carnitine contributed to a 47.6% and 46.3% increase (*p* = 0.044) in the Cu content, respectively, compared to the control group (Table 7).

There are no studies in the scientific literature on the effect of Bio-Mos and L-carnitine supplementation on the content of basic minerals in turkey muscles and liver; therefore, it is impossible to compare the findings of the present study with results obtained by other authors. Hence, it seems advisable to conduct further research to determine the optimal dose of these additives to be used in turkey nutrition.

### 3.4. Bone Quality Indicators

One of the most important problems occurring during turkey rearing is their bone strength, which may be aggravated by an inappropriate composition of the feed mixture and may exert a negative effect on the production performance [93]. Researchers have long investigated the impact of poultry feed additives that have the potential to increase bone strength parameters, improve bone geometrical features, reduce the risk of bone diseases, or increase the utilization and retention of minerals (including Ca and P) in animal organisms. These additives are e.g., active forms of vitamin D_3_ [94], phytases [95], organic acids [96], probiotic fructans [97], chelates [98], and diatomaceous earth [99]. However, the effect of Bio-Mos and L-carnitine on the physical, morphometric, and strength parameters and on the bone mineral composition has not been fully elucidated to date. There are only few literature data confirming the increase in the availability of minerals (including Ca) achieved through supplementation of poultry diets with Bio-Mos and L-carnitine. Similarly, it has rarely been reported that other minerals may have a positive effect on the physical, morphometric, and strength parameters and the bone mineral composition.

The present results showed that the thigh bones of the turkeys after the rearing period were normally developed, and no signs of cracks, fractures, or other damage were observed. The bone tissue strength is also dependent on the weight and length of the bone. One of the determinants of the development and mineralization of the skeletal system is static loads resulting from weight gain; therefore, the bone weight per 100 g of body weight was assessed as well. The addition of L-carnitine and Bio-Mos had an impact on the body weight of the turkey females only in the final rearing period, which was reflected in the weight of their femurs. Bone strength, mineralization, and histomorphometry are strongly correlated with changes in the body weight [100].

The addition of both L-carnitine and Bio-Mos to the mixture for turkey females significantly (*p* < 0.05) increased the weight (by on average 10%) and the length (by approx. 5.6%) of the femur compared to the bone parameters in the control group (Table 8). The study showed a positive effect of both supplements on the bone weight and length. There are many studies investigating the causes and frequency of skeletal abnormalities in poultry, but there are only few current reports on the effect of L-carnitine on bone quality (including weight and length) [101,102]. This ingredient is mainly added to the feed to determine its effect on animal growth [103], reduction of abdominal fat [104], or immune response [105]. Few studies have confirmed that L-carnitine not only increases bone weight [69,106] but also prevents rapid and permanent age-related loss of bone mass [31].

The turkey females receiving L-carnitine and Bio-Mos exhibited an altered spatial distribution of bone tissue, as the addition of these compounds to the feed mixtures significantly (*p* = 0.046) reduced the femur Wf/A parameter compared to its value in the group of turkeys receiving the standard diet. This, in turn, may have resulted in changes in the bone architecture reflected in the higher values of geometric and cortical parameters, especially in the Bio-Mos supplemented group (Table 8). The higher values of the A parameter, which has a significant impact on changes in bone structure, may have resulted from the higher values of the vertical external diameter and horizontal external diameter in the Bio-Mos addition variant and from the higher values of horizontal diameter in the case of cortical indices.

The architectural features of bones resulting from the spatial distribution and microarchitecture of bone material are one of the determinants of bone tissue strength. The analysis of the cortical parameters showed a significant (*p* < 0.05) effect of the nutritional Bio-Mos supplementation on CS and CI (Table 8). In comparison with the group receiving the standard feed, an 18% increase in the CS values and a 7.5% increase in CI were observed. As reported by Oliveira et al. [91], supplementation with mannans contributes to higher bone strength by increasing the absorption and accumulation of Ca in the organism. Similar properties were determined in the case of oligofructose [17], which is the second prebiotic after MOS used as an additive in poultry nutrition [107]. A study conducted by Świątkiewicz et al. [108], who analyzed the effect of prebiotic fructans (inulin and oligofructose) on the performance and bone characteristics in broiler chickens, did not show favorable results, whereas other researchers reported a potential beneficial effect of fructans. In another study, Świątkiewicz et al. [109] found a significant effect of oligofructose on tibia breaking strength and yielding load in laying hens. In turn, in a study conducted by Ortiz et al. [110], the addition of inulin to feed increased the content of crude ash and Ca in broiler tibias but had no effect on bone morphometric parameters. In a model study on ovariectomized rats, Zafar et al. [111] showed that the protective effect of fructans on bones was a result of increased Ca absorption and Ca balance in the organism, which resulted in enhanced bone mineralization and reduced bone turnover.

Changes occurring in bones throughout the lifetime are reflected in changes in mechanical parameters. Although bones are hard, they exhibit some plasticity and flexibility and react to continuous or repeated deformation forces by changing their structure [112]. The present study indicated that the introduction of L-carnitine and Bio-Mos into the mixtures for turkey females had a positive effect on the majority of the bone strength parameters (Table 8). However, the results of other authors’ studies on the effect of the use of Bio-Mos in feed did not confirm these results. The addition of 1 g/kg of MOS in the diet for forced molted and fully-fed laying hens had no effect on the mechanical properties of bones and the content of crude ash and minerals (Ca and P) in chicken bones [113]. The effect of Bio-Mos and L-carnitine used in the nutrition of the Big-6 turkeys on bone strength measured by the elastic force value, called the elastic point, after which the bone behaves like a plastic body (Wy), was determined in the present study. The addition of L-carnitine contributed to an increase in this parameter, while both additives caused an increase in the dy value, compared with the group fed the standard mixture. In comparison with its value in the bones of the control group, the dy parameter increased by 9% in the Bio-Mos-supplemented group and by approx. 15% in the L-carnitine variant. Additionally, bone mineralization reflected in the BI value was found to decline after the use of L-carnitine. In their study, Oliveira et al. [91] proved the positive effect of mannans on the bone mineralization process, which increased bone strength.

The bone stiffness parameter, defined as the Wy/dy, had the highest values in the group of turkeys receiving Bio-Mos. The lowest value of this parameter was determined in the group of turkeys supplemented with the standard complete mixture. However, the differences were not confirmed statistically. The lowest value of the Wf (720.2 N·mm) was obtained in the group of birds receiving L-carnitine in drinking water (Table 8). In turn, the use of Bio-Mos in the diet for turkeys significantly increased this parameter, compared to the Wf value in the bones from the group supplemented with L-carnitine. The analysis of bone strength revealed higher values of elasticity and fracture strength of femurs in the groups receiving the experimental additives. The beneficial effect of Bio-Mos and L-carnitine on the development of bones and their mechanical strength was greater than we expected, which suggests that the compounds may be promising nutritional additives supporting the development of the skeletal system in turkeys.

In a study conducted by Aydin et al. [33], the effect of L-carnitine on rats with advanced osteoporosis and femur fractures was evaluated. X-ray images showed significantly enhanced callus formation and fracture healing in groups treated with 50 mg/kg and 100 mg/kg L-carnitine. Additionally, bone mineral density (BMD) was significantly improved, while the serum levels of bone turnover markers (osteocalcin and osteopontin) and pro-inflammatory cytokines were reduced in the L-carnitine treated groups. The results showed that the L-carnitine supplementation improved femoral fracture healing. Another study [31] confirmed these findings. L-carnitine applied in the doses of 50 mg/kg and 100 mg/kg was able to restore BMD to the values measured in both ovariectomized and control animals. In ovariectomized rats, inflammation assessed by serum cytokine levels (TNF-α, IL-1β, and IL-6) intensified bone loss, as confirmed by the total femur BMD values. Both the lower and higher doses of L-carnitine were found to reduce bone loss significantly and improve inflammatory biomarkers. Positive effects of L-carnitine applied in rats were also reported by Ahmed et al. [114]. The aforementioned results may highlight the strong protective effect of L-carnitine in the treatment of osteoporosis and may lead to finding a new therapeutic target in other bone diseases in animals. Furthermore, L-carnitine may also prove effective as a first-choice dietary supplement in humans (female osteoporosis patients).

The content of crude ash in the femur of the turkey females, which is a good indicator of enhanced bone mineralization, was not dependent on the experimental additive used in the present study (Table 8). Similarly, the content of most minerals (Ca, Mg, Zn, and Cu) did not depend on the L-carnitine and Bio-Mos addition (Table 8). The P content in the femur was significantly higher in the Bio-Mos-supplemented group. Furthermore, the introduction of both additives resulted in a significant decrease in the K content in the femurs (by 10 and 15%, respectively), in comparison with the group fed the standard mixture. Although the absence of relationships of the content of mineral components (Ca, Mg, Zn, and Cu) with the additives was not statistically confirmed, the L-carnitine and/or Bio-Mos groups exhibited higher concentrations of minerals (except for Mg and Cu). This may indicate improved utilization of feed ingredients and an increased rate of mineral fraction apposition in the bones. The complex mechanism of the potential positive effect of prebiotics on the solubility, utilization, absorption, and bioavailability of minerals can be associated with such factors as increased amounts of fermentation products combined with an optimal intestinal environment, lower pathogen counts, healthy intestinal epithelium increased production of short-chain fatty acids, and improved intestinal integrity [115,116,117,118], which is essential for bone quality. In turn, the differences in the bone indices in the group of birds fed the prebiotic-supplemented diets may be associated with the fact that the effectiveness of these additives depends on many factors. These include the age of the birds, the total composition of the diet, the content of the prebiotic in the feed, and the animal production traits [113].

Since there are only few reports in the available literature on the effect of L-carnitine and Bio-Mos on all turkey bone parameters, it is not entirely possible to compare the present results with studies conducted by other authors. Therefore, the present results have some theoretical and practical values.

## 4. Conclusions

The results of the study suggest that the addition of L-carnitine and Bio-Mos in the feed for turkey broilers can modify the content of some macro and microelements in muscles and liver tissue and have a positive effect on weight gains in the final fattening period and on the composition of the breast muscles. Additionally, a positive effect of these feed additives on some femur quality indices was observed. It is therefore advisable to conduct further research focused on determination of the optimal level of supplementation and precise estimation of the potential benefits of L-carnitine and Bio-Mos supplementation in turkey nutrition, which can improve the rearing performance and welfare of these birds. However, a certain limitation of the research may be the way supplements are administered.

## Figures and Tables

**Table 1 animals-13-00770-t001:** Ingredient and nutrient content of the standard experimental diets (g 100 g^−1^, as fed-basic).

Ingredient	Diets		
	Grower 1 (6–9 Week)	Grower 2 (10–13 Week)	Finisher 1 (14–16 Week)
Maize, %	25.0	25.0	20.0
Wheat, %	30.6	36.8	56.6
Soybean, %	33.5	28.0	15.0
Meat and bone meal, %	5.00	5.00	4.00
Soya oil, %	2.00	2.00	1.20
Limestone, %	0.70	0.50	0.50
Cytromix Plus ^1^, %	0.20	0.20	0.20
Vitamin–mineral premix ^2^, %	3.00	2.50	2.50
Calculated nutrient composition, %			
Metabolizable energy, MJ‧kg^−1^	12.13	12.34	12.55
Crude protein, g kg^−1^	23.0	19.5	17.0
Lysine, g kg^−1^	1.45	1.25	1.05
Methionine+cysteine, g kg^−1^	0.95	0.85	0.75
Tryptophan, g kg^−1^	0.25	0.21	0.18
Threonine, g kg^−1^	0.92	0.79	0.67
Na, g kg^−1^	0.15	0.15	0.15

^1^ Cytromix Plus: citric acid, fumaric acid, phosphoric acid (62%). ^2^ Vitamin–mineral premix (per kg diet): vitamin A, 3,000,000 IU; vitamin D_3_, 900,000 IU; vitamin E, 10,000 mg; vitamin K_3_, 500 mg; vitamin B_1_, 700 mg; riboflavin, 2000 mg; vitamin B_6_, 1200 mg; vitamin B_12_, 6 mg; folic acid, 400 mg; biotin, 72 mg; niacin, 15,000 mg; choline, 120,000 mg; calcium pantothenicum, 4200 mg; Mn, 30,000 mg; Zn, 18,000 mg; Fe, 12,000 mg; Cu, 3000 mg; I, 200 mg; Se, 60 mg; Co, 40 mg; Ca, 15 g.

**Table 2 animals-13-00770-t002:** Content of mineral components in the feed mixtures for turkeys.

Mineral Components	Diets		
	Grower 1 (6–9 Week)	Grower 2 (10–13 Week)	Finisher 1 (14–16 Week)
Mg, g·kg^−1^	1.77	2.48	2.43
Ca, g·kg^−1^	8.58	7.77	7.67
K, g·kg^−1^	10.7	10.8	8.30
P, g·kg^−1^	10.4	10.9	9.89
Cu, mg·kg^−1^	22.1	34.8	16.6
Zn, mg·kg^−1^	171.6	131.3	123.7
Fe, mg·kg^−1^	304.7	327.3	345.4

**Table 3 animals-13-00770-t003:** Experimental design.

Experimental Factors	Groups		
	Control	L-Carnitine	Bio-Mos
Dose of Bio-Mos, % kg^−1^ mixture	-	-	0.5
Dose of L-carnitine, mL·L^−1^ H_2_O	-	0.83	-
Number of birds in the group	120	120	120
Number of birds slaughtered	12	12	12

**Table 4 animals-13-00770-t004:** Effect of L-carnitine and Bio-Mos on selected growth performance parameters and organ weight in female turkeys ^1^.

Item		Control	L-Carnitine	Bio-Mos	*p*-Value	SEM
Body weight, kg	6 weeks	1.74	1.76	1.78	0.061	0.025
	9 weeks	3.23	3.58	3.69	0.057	0.247
	15 weeks	8.51	8.77	8.98	0.058	0.321
	16 weeks	9.12 ^b^	9.67 ^a^	9.68 ^a^	0.047	0.273
Gain, kg (6–16 weeks of life)		7.83	7.90	7.91	0.054	0.054
FCR, kg‧kg^−1^ (6–16 weeks of life)		2.68	2.69	2.61	0.051	0.045
Organ weight, g						
Breast muscle		1657	1884	1875	0.057	0.036
Thigh muscle		658	699	687	0.051	0.023
Liver		154	147	143	0.087	0.021

^1^ Data represent the mean of 12 birds per treatment. Control—diet without L-carnitine and mannan oligosaccharides (Bio-Mos); L-carnitine—dose 0.83 mL·L^−1^ H_2_O; Bio-Mos—0.5%/kg mixture; FCR—feed conversion ratio; SEM—standard error of the mean. ^a,b^—means within a row with different superscripts differ significantly at *p* < 0.05.

**Table 5 animals-13-00770-t005:** Content of basic nutrients and mineral elements in female turkey’s breast muscle ^1^.

Item	Control	L-Carnitine	Bio-Mos	*p*-Value	SEM
Basic nutrients, g 100 g^−1^					
Dry matter	27.2	27.8	26.9	0.058	0.235
Crude ash	3.28 ^a^	2.79 ^b^	2.93 ^b^	0.043	0.112
Crude protein	26.1	25.9	24.9	0.058	0.153
Crude fat	0.93 ^a^	0.69 ^c^	0.71 ^b^	0.029	0.031
Mineral elements					
Mg, g·kg^−1^	0.43	0.47	0.45	0.064	0.005
Ca, g·kg^−1^	0.06	0.08	0.08	0.074	0.001
K, g·kg^−1^	2.58	2.67	2.59	0.059	0.031
Na, g·kg^−1^	0.45 ^ab^	0.43 ^b^	0.48 ^a^	0.049	0.053
Fe, mg·kg^−1^	6.71	7.28	6.97	0.058	0.951
Cu, mg·kg^−1^	1.68 ^b^	1.81 ^ab^	2.64 ^a^	0.038	0.124
Zn, mg·kg^−1^	14.9	15.4	13.9	0.087	0.313

^1^ Data represent the mean of 12 birds per treatment. Control—diet without L-carnitine and mannan oligosaccharides (Bio-Mos); L-carnitine—dose 0.83 mL·L^−1^ H_2_O; Bio-Mos—0.5%/kg mixture; SEM—standard error of the mean. a,b,c—means within a row with different superscripts differ significantly at *p* < 0.05.

**Table 6 animals-13-00770-t006:** Content of basic nutrients and mineral elements in female turkey’s thigh muscle ^1^.

Item	Control	L-Carnitine	Bio-Mos	*p*-Value	SEM
Basic nutrients (g 100 g^−1^)					
Dry matter	23.9	24.0	24.5	0.073	0.214
Crude ash	2.11	2.18	2.15	0.068	0.082
Crude protein	21.9	22.0	21.9	0.088	0.213
Crude fat	1.73	1.83	1.85	0.074	0.032
Mineral elements					
Mg, g·kg^−1^	0.12	0.11	0.10	0.077	0.031
Ca, g·kg^−1^	0.032 ^b^	0.039 ^a^	0.035 ^ab^	0.047	0.002
K, g·kg^−1^	1.82	1.68	1.69	0.056	0.034
Na, g·kg^−1^	0.58	0.57	0.52	0.084	0.013
Fe, mg·kg^−1^	10.3 ^b^	8.24 ^c^	12.9 ^a^	0.039	0.431
Cu, mg·kg^−1^	1.28 ^b^	1.99 ^a^	0.61 ^c^	0.044	0.083
Zn, mg·kg^−1^	18.9	18.3	20.0	0.051	0.045

^1^ Data represent the mean of 12 birds per treatment. Control—diet without L-carnitine and mannan oligosaccharides (Bio-Mos); L-carnitine—dose 0.83 mL·L^−1^ H_2_O; Bio-Mos—0.5%/kg mixture; SEM—standard error of the mean. a,b,c—means within a row with different superscripts differ significantly at *p* < 0.05.

**Table 7 animals-13-00770-t007:** Content of basic nutrients and mineral elements in female turkey’s liver ^1^.

Item	Control	L-Carnitine	Bio-Mos	*p*-Value	SEM
Basic nutrients (g 100 g^−1^)					
Dry matter	32.1	31.9	31.3	0.057	0.213
Crude ash	5.98	4.92	5.18	0.062	0.124
Crude protein	17.9	18.5	18.5	0.057	0.215
Crude fat	3.81 ^a^	2.95 ^b^	3.32 ^c^	0.027	0.098
Mineral elements					
Mg, g·kg^−1^	0.34	0.33	0.32	0.053	0.013
Ca, g·kg^−1^	0.15 ^a^	0.11 ^ab^	0.06 ^b^	0.037	0.008
K, g·kg^−1^	2.25	2.39	2.35	0.059	0.07
Na, g·kg^−1^	1.12	1.05	1.13	0.067	0.315
Fe, mg·kg^−1^	93.1 ^b^	116.4 ^ab^	128.4 ^a^	0.043	7.442
Cu, mg·kg^−1^	4.93 ^b^	9.18 ^a^	9.41 ^a^	0.044	0.551
Zn, mg·kg^−1^	47.3	49.6	48.3	0.079	2.482

^1^ Data represent the mean of 12 birds per treatment. Control—diet without L-carnitine and mannan oligosaccharides (Bio-Mos); L-carnitine—dose 0.83 mL·L^−1^ H_2_O; Bio-Mos—0.5%/kg mixture; SEM—standard error of the mean. a,b,c—means within a row with different superscripts differ significantly at *p* < 0.05.

**Table 8 animals-13-00770-t008:** Physical, morphometric parameters, strength parameters, and mineral composition of crude ash in female turkey’s femur bones ^1^.

Item	Control	L-Carnitine	Bio-Mos	*p*-Value	SEM
Physical parameters					
Bone weight					
g	36.3 ^b^	39.7 ^a^	39.9 ^a^	0.035	0.676
g/100 g	0.47	0.40	0.45	0.160	0.009
Length, mm	117 ^b^	122 ^a^	125 ^a^	0.477	0.009
Perimeter, mm	47.2	48.3	49.1	0.740	1.038
Geometric features					
Ix, mm^4^	948.6	953.2	997.6	0.890	43.25
A, mm^2^	50.1 ^b^	53.4 ^ab^	59.8 ^a^	0.010	1.440
MRWT	0.25	0.29	0.29	0.240	0.011
Cortical indices					
CLT, mm	3.53	3.55	3.75	0.617	0.096
CS, m^2^	81.3 ^c^	90.6 ^b^	95.8 ^a^	0.045	2.696
CI, %	34.6 ^b^	33.9 ^b^	37.2 ^a^	0.038	0.757
CSI, %	41.1	41.0	42.4	0.791	0.876
Strength parameters					
Wy, N‧mm	199 ^b^	209 ^a^	203 ^ab^	0.048	2.995
dy, mm	1.89 ^b^	2.13 ^a^	2.02 ^a^	0.038	0.051
Wf, N·mm	808 ^a^	720 ^b^	826 ^a^	0.035	31.17
Wy/dy, N‧mm·mm^−1^	99.0	99.2	101.1	0.934	2.201
Wf/A, N·mm·mm^−2^	15.9 ^a^	13.4 ^b^	13.8 ^b^	0.046	0.508
BI, mg·mm^−1^	295 ^b^	348 ^a^	314 ^ab^	0.093	10.22
Mineral composition					
Crude ash, %	27.4	28.3	28.6	0.067	0.217
P, g‧kg^−1^	95.3 ^b^	99.5 ^ab^	101.5 ^a^	0.009	0.908
Mg, g·kg^−1^	9.03	8.75	8.93	0.281	0.072
Ca, g·kg^−1^	161	165	167	0.405	1.772
K, g·kg^−1^	7.33 ^a^	6.63 ^b^	6.25 ^b^	0.002	0.131
Cu, mg·kg^−1^	14.8	14.6	14.5	0.377	0.079
Zn, mg·kg^−1^	304	308	306	0.441	3.688

^1^ Data represent the mean of 12 birds per treatment. Control—diet without L-carnitine and mannan oligosaccharides (Bio-Mos); L-carnitine—dose 0.83 mL·L^−1^ H_2_O; Bio-Mos—0.5%/kg mixture; SEM—standard error of the mean; CLT—thickness of cortical layer; CS—cortical surface; CI—cortical index; CSI—cortical surface index; Wy—yielding load; dy—yielding deformation; Wf—maximum force moment; Wy/dy—load-to-deformation ratio; Wf/A—bending point resistance; BI—density index; Ix—second moment of inertia; A—cross-sectional area; MRWT—mean relative wall thickness. a,b,c—means within a row with different superscripts differ significantly at *p* < 0.05.

## Data Availability

Not applicable.

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
