# Peer review of "Effects of Dietary Supplementation of L-Carnitine and Mannan-Oligosaccharides on Growth Performance, Selected Carcass Traits, Content of Basic and Mineral Components in Liver and Muscle Tissues, and Bone Quality in Turkeys"

_animals, 2023, doi:10.3390/ani13040770_

Round 1
Reviewer 1 Report (New Reviewer)
This study aims to determine the effect of L-carnitine and Bio-Mos on the given given parameters in turkeys. The experimental design and and aims are fair, just minor changes are recommended;
Introduction - please justify the use of L-carnitine with Bio-Mos together, by stating components that might supplement each other, e.g name elements or compounds absent in L-carnitine but present Bio-Mos
Line 121 - Please include the type of house used for the experiment and the name of the research facility were the feeding trial took place.
Line 161 - Please use a better word for dead, e.g after slaughter weight or post slaughter weight
Line 156 - Please expand or define the methods on how growth performance was collected or determined (gain, FCR and feed intake)
Author Response
Response to Reviewer 1 Comments
Point 1: Introduction - please justify the use of L-carnitine with Bio-Mos together, by stating components that might supplement each other, e.g name elements or compounds absent in L-carnitine but present Bio-Mos.
Response 1: Explanatory paragraph added to the Introduction (L: 131-141):
The combination of Bio-Mos and L-carnitine would make this combination much more beneficial than the application of single ingredients. Bio-Mos is a prebiotic, and therefore it stimulates the development of a beneficial intestinal microbiome, improves the morphological structure of the intestinal epithelium, protects enterocytes, and also stimulates the immune system located largely in the digestive tract. On the other hand, L-carnitine facilitates the penetration of fatty acids into cells, thanks to which they can be used to improve cell energy and reduce glycogen and fat reserves in the carcass. This entails a measurable health effect and improves weight gain. In addition, L-carnitine stimulates the distribution of important short-chain fatty acids, e.g. butyric acid, which is a product in the enterocytes of the intestinal microbiota. This, in turn, can stimulate the work of intestinal enteroendocrine cells and regulate their hormonal activity [Lu et al., 2021; Silva et al., 2020; García-Cabrerizo et al., 2021].
Point 2: Line 121 - Please include the type of house used for the experiment and the name of the research facility were the feeding trial took place.
Response 2: The research was carried out in a livestock building intended for industrial farming of turkeys. At the time of the experiment, it was a turkey farm and is now a laying hen farm. Facility address: 21-007 Mełgiew, ul. Piasecka 49A, Poland.
Point 3: Line 161 - Please use a better word for dead, e.g after slaughter weight or post slaughter weight.
Response 3: The sentence has been changed to: “Post slaughter weight of turkey females was included in the calculation of the FCR.”, L: 219-220.
Point 4: Line 156 - Please expand or define the methods on how growth performance was collected or determined (gain, FCR and feed intake).
Response 4: Explanatory paragraph added to the Materials and Methods (L: 214-219):
A 3-phase feeding programme (week 6–9, 10–13 and 14–16) was used during the study. At 6, 9, 13 and 16 week the turkey hens were weighed and feedintake was recorded. The body weight gain of birdsand feed conversion ratio were calculated for eachfeeding period. Mortality rates were recorded daily and the body weights of all dead birds were used to adjust for average daily gain, average daily feed intake and feed conversion ratio.
Thank you for your helpful comments that will help to improve this paper.

Reviewer 2 Report (New Reviewer)
The impact of feed additives on selected production indices and the quality of poultry carcasses is still valid—as well as the effect of feed additives on bone tissue. Therefore, the presented research is cognitive and practical.
TITLE: corresponds to the work’s content.
ABSTRACTS: correct.
KEYWORDS: correct.
OBJECTIVES: they were properly established and were fulfilled.
INTRODUCTION: correct.
· Please complete lines 110-112 in the sentence "optimal nutritional value" for whom? consumer?
· At the end of the Introduction, he will separate the objectives into a new paragraph.
MATERIAL AND METHODS: Methods and calculation methods are properly described. Statistical methods correctly adjusted.
· Lines 138-139 specify the method used to analyze the minerals in the feed listed in Table 2
· Table 3 shows 100 birds per group, and rows 125-127 show 120 birds. Please explain.
· In lines 163-164, please quote the normative acts concerning the procedures for slaughtering turkeys.
· Lines 171-172, whether marked crude protein or total nitrogen, measured with the Kjeldahl method, converted (a conversion factor 6.25) into an amount of crude protein (%)?
· Lines 183 please complete how was the accuracy of the chemical determinations validated (certified reference material).
RESULTS AND DISCUSSION:
· Please explain why it was decided to study females? In the discussion, it is worth noting the influence of hormonal balance in turkey females on lipid metabolism and bone tissue.
· In the headings of Tables 4 to 8, please enter the sex of the turkeys.
· Table 4 should not show absolute organ weights, but relative weights (g x 100g-1 of body weight). In its current form, table 4 shows that the liver weighed less than 2 g.
· How do the authors explain the decrease in ash content in breast muscles under the influence of the supplementation (line 290-291)?
· Will the consumption of liver (as an offal product) be beneficial for the consumer, taking into account the increase in its content of iron and copper and the decrease in fat (Table 6) under the influence of the supplementation used? Please respond to this.
· In the authors' opinion, will the increase in the body weight of turkeys ( 5%) cover the cost of including the additives in their diet? Will such turkey diet supplementation be economically viable for the producer?
· Please add in the last paragraph what was the limitation of the research.
CONCLUSIONS: Properly formed.
REFERENCES: I propose to reduce the cited literature to the necessary items.
Author Response
Response to Reviewer 2 Comments
Point 1: Please complete lines 110-112 in the sentence "optimal nutritional value" for whom? consumer?
Response 1: Clarified according to the Reviewer's suggestion, L: 146-147.
Point 2: At the end of the Introduction, he will separate the objectives into a new paragraph.
Response 2: The objective is separated into a new paragraph, L: 151-155.
Point 3: Lines 138-139 specify the method used to analyze the minerals in the feed listed in Table 2.
Response 3: The method has been specified, L: 176-194.
Point 4: Table 3 shows 100 birds per group, and rows 125-127 show 120 birds. Please explain.
Response 4: Thank you for the vigilance, Reviewer. By mistake, instead of 120 birds, 100 birds were given. Corrected in Table 3.
Point 5: In lines 163-164, please quote the normative acts concerning the procedures for slaughtering turkeys.
Response 5: The normative acts added to the Materials and Methods, L: 222-226.
Point 6: Lines 171-172, whether marked crude protein or total nitrogen, measured with the Kjeldahl method, converted (a conversion factor 6.25) into an amount of crude protein (%)?
Response 6: In the biological material, total nitrogen was determined using the Kjeldahl method, on the basis of which the amount of crude protein, with using the factor 6.25.
Point 7: Lines 183 please complete how was the accuracy of the chemical determinations validated (certified reference material).
Response 7: Certified reference material and the content of the analyzed minerals have been added to the text of the paper, L: 246-248.
Point 8: Please explain why it was decided to study females? In the discussion, it is worth noting the influence of hormonal balance in turkey females on lipid metabolism and bone tissue.
Response 8: Turkey hens were used in the experiment due to the fact that it is our permanent research material, to which we have access and for which the henhouses were adapted. Moreover, rearing of a turkey hens' is shorter than that of a turkey, which translates into the economic aspect of the experiment.
Of course, it is hard to deny that estrogens have a significant impact on the lipid metabolism and the qualitative characteristics of bone tissue, however, the studies concerned turkeys aged from 6 to 16 weeks (before reaching puberty). This seems to minimize the likelihood of excessive carcass fatness (with standardized nutrition). On the other hand, the deterioration of the characteristics of bone tissue concerns individuals with estrogen deficiency (Reference: 106.).
Point 9: In the headings of Tables 4 to 8, please enter the sex of the turkeys.
Response 9: Sex of the turkeys (female turkeys) added to the headings of Tables 4-8.
Point 10: Table 4 should not show absolute organ weights, but relative weights (g x 100g-1 of body weight). In its current form, table 4 shows that the liver weighed less than 2 g.
Response 10: Thank you for the vigilance, Reviewer. Of course, in Table 4, the weight of the liver was changed, giving it in grams.
Point 11: How do the authors explain the decrease in ash content in breast muscles under the influence of the supplementation (line 290-291)?
Response 11: It is possible that the decrease in crude ash content in the breast muscle was the result of a lower content of some macroelements, but this was not the subject of our study.
Point 12: Will the consumption of liver (as an offal product) be beneficial for the consumer, taking into account the increase in its content of iron and copper and the decrease in fat (Table 6) under the influence of the supplementation used? Please respond to this.
Response 12: Liver consumption is recommended in the case of anemia, so the increased content of Fe and Cu seems to be beneficial in this aspect. On the other hand, the lower fat content results in lower calorific value, which in some situations can be a positive advantage.
Point 13: In the authors' opinion, will the increase in the body weight of turkeys (5%) cover the cost of including the additives in their diet? Will such turkey diet supplementation be economically viable for the producer?
Response 13: We didn’t achieve impressive weight gains in our experiment, nor did we achieve a reduction in FCR in connection to control group. Indeed, looking at the economic aspect, such supplementation does not bring measurable benefits in the form of the effectiveness of rearing turkeys. In large-scale breeding, of course, this would not have an economic impact. We agree with the Reviewer. Nevertheless, our task was also to assess the impact of Bio-Mos and L-carnitine on the improvement of bone mechanics, i.e. on physical, morphometric and strength parameters as well as bone mineral composition, which in our opinion has not been fully explained in the literature so far. We consider this aspect of our research satisfactory.
For we have found out, Bio-Mos was found to favorably increase bone cross-section and density; it also improved bone strength and cortical indices. It can therefore be assumed that this supplement had a better effect than L-carnitine on the use and incorporation of mineral elements into bone tissue, thus improving bone strength and nutrition. Skeletal strength is a very important aspect in large herd farming, as numerous bone fractures are often the cause of turkey leg injuries and mortality, especially in the final rearing period. Looking at the cognitive aspect of the research and the potential application, it can be concluded that this effect has been achieved.
Point 14: Please add in the last paragraph what was the limitation of the research.
Response 14: Limitation of the research was added to the Conclusions, L: 741-742.
Point 15: I propose to reduce the cited literature to the necessary items.
Response 15: Due to the multifaceted nature of the research (assessment of L-carnitine and Bio-Mos on the results of production, carcass mineral composition and bone quality), it seems that the amount of literature used is justified. Hence, it is difficult to give up without detriment to the analyzed subject matter.
Thank you for your helpful comments that will help to improve this paper.

Reviewer 3 Report (Previous Reviewer 2)
Authors evaluate the effect of L-carnitine and Bio-Mos administration on selected production performance, slaughter parameters, elemental and mineral content of liver, breast and thigh muscles, and physical, morphometric, strength and bone mineral composition parameters of turkeys..
The title indicates the aim of the manuscript and the abstract is well written. It clearly indicates the work objective, methodology and result of the study.
The introduction is also well written.
The objectives of the study are of interest and are in line with the scope of the journal.
The manuscript is well organized.
The conclusions are consistent with the evidence and arguments presented.
The reference is appropriate.
Some suggestions are required, as follows:
Line 51-52: Add reference
Line 567: Delete The study was conducted in accordance with the Declaration of Helsinki.
In my opinion, the manuscript could be accepted for publication after minor revision.
Author Response
Response to Reviewer 3 Comments
Point 1: Line 51-52: Add reference.
Response 1: Two references added:
Henchion, M.; Zimmermann, J. Animal food products: policy, market and social issues and their influence on demand and supply of meat. Proc. Nutr. Soc. 2021, 80, 252–263.
OECD FAO Agricultural Outlook 2021-2030. 2021. Available online: https://reliefweb.int/attachments/237350c5-87ab-3e2a-acb5-45e72071993f/cb5332en.pdf (accessed on 14 February 2023), pp. 163–177.
Point 2: Line 567: Delete The study was conducted in accordance with the Declaration of Helsinki.
Response 2: The sentence deleted.
Thank you for your helpful comments that will help to improve this paper.

This manuscript is a resubmission of an earlier submission. The following is a list of the peer review reports and author responses from that submission.
Round 1
Reviewer 1 Report
At present, there have been many studies on the effects of dietary copper supplementation on animal growth and health, and I think this work is not innovative enough to meet the requirements of publication in the journal of Animals.
Reviewer 2 Report
The authors extended their research on the effects of Cu-Gly administration on rat body, heart, liver and kidney weight and blood plasma and basal vital organs contents of Cu, Fe, and Zn.
The title indicates the aim of the manuscript and the abstract clearly indicates the work objective, methodology and result of the study. The methodology is well articulated and the description is well made.
Manuscript is well-structured and scientifically sound therefore. The conclusions are consistent with the evidence and arguments presented.
I have a suggestions
Given that in this procedure the animals were euthanized (line 147), the authorization by competent authority of the is lacking based on Directive 2010/63/EU of the European Parliament and of the Council of 22 September 2010 on the protection of animals used for scientific purposes. It is present only the approval of Local Ethics Committee. Please, insert the approval number for this study involving animals.
I support the publication of this manuscript after the abovesaid minor revision.
Reviewer 3 Report
The authors investigated the effect of the source of copper (sulfate & chelate) in the diet on body weight and the content of Cu, Fe, and Zn in selected basal vital organs of adult rats. They designed five treatments to test their hypothesis. This manuscript (MS) was clearly written and easy to understand. However, some major issues significantly compromised the quality of this MS.
Major comments:
- First, the manuscript needs to be edited by a native English speaker to improve the language of the MS and fix errors.
- Some part of the introduction and discussion is repetitive, and the authors could not organise an understandable story and structure in the introduction. A deep revision in the introduction is required.
· “had an effect on the liver weight in both analysed periods”. Please avoid this style of writing in the MS and revise the MS from this point.
However, I have touched on some more points that can contribute to the improvement of this MS.
Minor comments
Abstract
· Line 19, make sure you define the abbreviations for the first time in the MS.
· Line 21, this sentence is not clear and how it is relative to methionine or lysine.
· Line 36-39, please revise this sentence.
· It is better to start the abstract with a sentence about why you did this study.
· Here and elsewhere, report P uppercase and italic (P<0.05).
· Throughout the MS, if there is no significant difference, there is no need to report P-value.
· Please reorder the keywords alphabetically and capitalise each word.
· Please write the abstract more numerically about the results. You can do it by adding their numbers in parentheses.
Introduction:
• Well-developed introduction and included a clear fellow and relevant points.
· Line 48, please add the abbreviation of iron and zinc.
· Line 48-50, what is the difference between growing and adult? Both look the same.
· Line 57, delete “Moreover”.
· Line 36, you defined copper and other elements earlier. Therefore, please be consistent with using their abbreviations (Cu…)
· Line 70-73, please check it; you already mentioned this part earlier.
· Line 86-92, you mentioned some parts of that earlier in the introduction; please revise it.
· Line 102-104, I could not understand the relation between metals and methionine or lysine chelates. The authors should clear this point somewhere in the MS.
· Line 104, “glycine chelate” should be explained well.
· Here and throughout the MS, please first mention the common name plus the scientific name, and for the rest of the MS, just report the common name.
· Please update the introduction with recent works as many studies are available from the last two years, which were not included in this section.
· Please mention the novelty of your work in the last paragraph of the introduction.
Material and methods
· Well-organised section. Clear fellow and all required details were provided.
· For each analysis, please clarify how many animals were taken.
· Jodine?? Table 1, and which diet???
· Table 2, please add the level of CuSO4 in the diet (5 mg/kg body weight per day from sulfate (CuSO4). The requirement of each animal is around 1 mg as the weight of the animal is aprox 200 grams. How do you calculate the dosage of diets so that with feeding 0.025 mg Cu /1l H2O you could provide 1 mg. Please explain clearly these calculations as I think some mistake has occurred.
· Table 3, in where diet or tissue? Please explain in the title of the tables.
Results
· Well-written section; all necessary things have been covered but need to be more numeric.
· Line 180-182, please make it clear was significant or not. Please check this point for all results and if the change is significant should be clear.
· Table 4, please explain how you measured Cu consumption, mg/l and add how much is required based on “mg/l”.
· Line 196-197, please avoid this style of writing. How affected? “the addition of Cu as Cu-Gly had an effect on the liver weight in both analysed periods”. Please revise the results section from this point
· Higher liver weight or HSI is not a good sign of health. Please explain well in discussion what is the reason and whether this increase is in a standard range?.
· Please combine tables 5 and 6.
Discussion
• Line 252, is not!!!! Please provide a reference if you think so.
• Line 254, please avoid using some extreme verbs, adverbs, and adjectives in scientific writing. Please revise the MS.
· Some parts of the discussion are better updated with research in 2021 and 2022 as they refer to some old references. Please update the discussion with the latest studies as much as possible.
· Although you wrote this section well, you can still improve it by answering these questions and annotating them in the discussion section. Why were these results observed? Discuss more possible reasons.
•
· The conclusion needs to be revised and more comprehensive concepts should be added there.
Tables and Figures
• Please explain a little bit about your experimental diets, per each Table and Figure. Each Table and figure should represent enough information separately from the text.
• Double-check the units and titles of all Tables.
• Please mention in the footnote of all Tables which kind of statistical method you used for comparing the means.
When revising your manuscript, please consider all issues mentioned in the reviewers' comments carefully with clear outlines for every change made in response to their comments including suitable rebuttals for any comments you deem inappropriate. Please itemise your response to each review comment, and highlight the revised at re-submission.
Best regards